

# CONTINUOUS VERTICAL AEROSOL PROFILING WITH A MULTI-WAVELENGTH RAMAN POLARIZATION LIDAR OVER THE PEARL RIVER DELTA, CHINA

Birgit Heese[1], Holger Baars[1], Stephanie Bohlmann[1], Dietrich Althausen[1], and Ruru Deng[2]

[1]Leibniz Institute for Tropospheric Research (TROPOS), Permoserstrasse 15, 04318 Leipzig, Germany
[2]Sun Yat-sen University, Guangzhou, China

*Correspondence to:* Birgit Heese
(heese@tropos.de)

**Abstract.**

A dataset of particle optical properties of the highly polluted atmosphere over the Pearl River Delta, Guangzhou, China, is presented in this paper. The data were derived from the measurements of a multi-wavelengths Raman and depolarization lidar Polly$^{XT}$ and a co-located AERONET sun photometer. The measurement campaign was conducted from November 2011 to

mid June 2012. These are the first Raman lidar measurements in the PRD that were lasting for several months.

A mean value of aerosol optical depth observed by the sun photometer of 0.54 ± 0.33 was observed in the polluted atmosphere over this megacity for the whole measurement period. The lidar profiles frequently show lofted aerosol layers which reach up to altitudes of 2 to 3 km and especially in spring season up to even 5 km. These layers contain between 12 and 56% of the total AOD, with the highest values in spring. The aerosol in these lofted layers were classified by their optical properties.

The observed lidar ratio values are in the range from 30 to 80 sr with a mean value of 48.0 sr ± 10.7 sr. The linear particle depolarization ratio at 532 nm lied mostly below 5% with a mean value of 3.6 ± 3.7. The majority of the Ångström exponents were observed between 1 and 1.5 indicating a coarse mode in addition to the fine mode in the particle size distribution.

These results reveal that mainly urban pollution particles mixed with particles arisen from biomass and industrial burning are present in the atmosphere above the Pearl River Delta. Trajectory analyses show that these pollution mixtures arise mainly

from local and regional sources.

## 1   Introduction

The Pearl River Delta (PRD) in the South-East of China is one of the largest urbanized areas in the world. High population density and a very strong economy leads to an almost permanent high aerosol load in the whole area around the city of Guangzhou in the PRD. The consequences for the geographical development, people's health, and atmospheric pollution were

studied in the frame of the German project "Megacities – Megachallenges". The atmosphere over the PRD is influenced by high urban and industrial activity but is also affected by the vicinity of the sea. Hence the predominant atmospheric particles



expected to be found in this area are a mixture of different aerosol types like urban haze, burning products from traffic and industry, and also sea-salt particles.

The visibility in Guangzhou has significantly decreased during the last four decades. Since 1972 the number of days with low visibility has increased from a few days per years to about 100-150 days per year from 1980 to 2006 (Deng et al., 2008). The authors could relate the low visibility to the increasing particle concentration observed by in-situ particle measurements during a case study in November 2005, where both high and low visibility episodes occurred. They found that 70% of the visibility is reduced by small scattering particles, and 20% by absorbing particles. A comprehensive overview over the air pollution at ground level since 1990 in megacities in China is given in Chan and Yao (2008). They found that particles arisen from traffic, industry, wood and coal burning are the major pollutants most of the time in the PRD, and can causes high pollution episodes and low visibility. This type of in-situ studies have revealed valuable information about the particle types and concentration measured at ground level that contribute to the severe air conditions in the PRD and other magacities. But how is the vertical distribution of these particles?

Only a limited number of vertically resolved aerosol observations are available over South-East China until the beginning of this century. The Asian dust lidar network was established in the late 1990 (Murayama et al., 2001) with lidar stations mainly in Japan and only occasionally in China. For example, a short-term study was conducted in July 2006 during the PRD2006 campaign (Sugimoto et al., 2009). Here, a two-wavelength, backscatter and polarization lidar from the National Institute for Environmental Studies (NIES) in Japan was used. Two typhoon-driven flow episodes of northern air, periods of accumulation of air pollution within the PRD area, and three cases of lofted layers above the PBL were observed. This lidar was staying in Guangzhou until March 2009 and was measuring in Guangzhou for a long-term study on seasonal aerosol variations (Hara et al., 2011).

First Raman lidar measurements in the PRD were carried out by TROPOS during a one-month intensive field campaign in Xinken in October 2004 (Tesche et al., 2007). The lidar used was the prototype of the mobile Raman lidar Polly with just one wavelength and two detection channels. High levels of aerosol load and the presence of lofted aerosol layers was observed during the entire period (Ansmann et al., 2005).

In November and December 2009, as part of the Megacities project, a first short field campaign took place in Zhongshan in the southern part of the PRD. During this campaign a Raman lidar of the Anhui Institute of Optics and Fine Mechanics, Hefei, China and a sun photometer from TROPOS were deployed. To contrast the different aerosol conditions, two significant events of moderate and hazy pollution were characterized in detail by Chen at al. (2014). To investigate the aerosol conditions of this highly polluted area over a longer time period and to study the inter-seasonal differences, long-term observations in the PRD were realised in the frame of the Megacities project. The campaign was performed from Autumn 2011until summer 2012 in Guangzhou. The multi-wavelength Raman lidar Polly$^{XT}$ (Althausen et al., 2009) with depolarization capabilities was used for the characterization of the aerosol types over the PRD. The obtained results are presented in this paper. To our knowledge, this is the first time that continuous Raman lidar observations have been performed in the PRD for more than half a year. This provides a unique data set of the vertical aerosol distribution including the characterization of the optical properties in this area.



In the next section the campaign details are given, the instrumental set-up is described, and the climatic conditions are outlined. In section 3 a seasonal overview over the lidar and sun photometer measurements results are given and a case study of particularly high aerosol contend in the vertical profile is presented. In section 4 the layered structure of the aerosol is analyzed, the aerosol is classified by lidar optical properties, and the origin of the observed aerosol is examined by trajectory

cluster analysis. A statistical analysis of the measured optical properties is presented as well, and finally, a conclusion is given.

## 2   Experiment

### 2.1   Field site

The measurements for the vertically-resolved aerosol characterization were taken by a multi-wavelength Raman lidar of type Polly$^{XT}$ (Althausen et al., 2009; Engelmann at al., 2016) of TROPOS and a dual-polar sun photometer of type CE-318dp

(Cimel) from AERONET (Holben et al., 1998). Both instruments were deployed on the roof-top of a laboratory building on the East-Campus of the Sun Yat-sen University of Guangzhou (23°04'08"N, 113°22'52"E, 23.5 m above sea level), where the meteorological education at the University takes place. The lidar was deployed on a roof terrace on the last floor of the building with easy access from the adjacent laboratories. The sun photometer was deployed on the flat roof-top of the building with a 360° panoramic sight undisturbed by obstacles. The sun photometer measurements started in the end of October 2011 and

lasted until the beginning of July 2012. The lidar was running in a continuous 24/7 measuring mode from the beginning of November 2011 until mid June 2012 - only interrupted by rain periods. In the case of rain, a rain sensor is detecting the falling rain drops and the system is closing down the measurements immediately.

The optical properties measured by the lidar are the particle backscatter coefficient at 355 nm, 532 nm, and 1064 nm, the particle extinction coefficient at 355 nm and 532 nm, and the linear depolarization ratio at 532 nm. For the determination of

the particle backscatter coefficient and particle extinction coefficient the Raman method (Ansmann et al., 1992) was applied. The linear total or volume depolarization ratio is including molecular depolarisation effects. The linear particle depolarization ratio was calculated using the 90° calibration method described in detail by Freudenthaler et al. (2009). Further derived optical properties are the extinction-to-backscatter-coefficient ratio, also called lidar ratio, the linear particle depolarization ratio, the aerosol optical depth (AOD), and the respective backscatter and extinction related Ångström exponents.

These properties are used to identify the type of aerosol that was observed. While the particle backscatter coefficient indicates the present amount of particles, depending on their scattering abilities, the particle extinction coefficient also relates to the absorption abilities of the particles. The lidar ratio, the ratio between these coefficients, is dependent of the particle type, not on the amount. Also the particle depolarisation ratio is typical for the particle type and helps distinguishing between spherical and non-spherical particles. The Ångström exponents describe the wavelength dependence of the respective backscatter and

extinction coefficients and are dependent on the size distribution of the particles.

The lidar data were analysed as follows: The data were visually cloud screened using the automatically produced quick-look images on the PollyNET-webpage: www.polly.tropos.de. Then, these data were evaluated manually in intervals of 2 h to 3 h to obtain comparable profiles for a statistical analysis. To not over-represent long lasting cloud free periods and unchanged





aerosol conditions on the same day, the number of considered profiles per day for such periods was reduced to a maximum of 4.

The lidar data presented here are without any overlap correction. The overlap function could not be calculated due to permanently high aerosol load in the atmosphere over the PRD. However, to be able to calculate the AOD from the lidar profiles, the Raman backscatter profiles were fitted to the Raman extinction profiles in the lower part of the profile. Then, the lowermost extinction value at about 150 m height was extrapolated to the ground-level.

The sun photometer measures the direct and indirect Sun radiation at nine wavelengths from 340 nm to 1640 nm every 15 min when the Sun is visible. In addition, with this dual-polar instrument, the radiances at three different depolarization directions are measured by means of a second filter wheel (Li et al., 2009). However, the data used in this study are derived solely from direct Sun measurements. The products used are the AOD and the Ångström coefficients derived by AERONET available from the AERONET-webpage: aeronet.gsfc.nasa.gov.

## 2.2  Atmospheric conditions

The atmospheric conditions of the PRD are controlled by a subtropical climate characterized by warm winters and hot and humid summers. The monthly mean temperatures are coldest in January: 13.9° C and warmest in July: 28.8°C. The annual mean temperature is 22.4 °C. The weather is influenced by the Asian monsoon circulation. The main wind direction turns with the dislocation of the Inter Tropical Convergence Zone (ITCZ) from north and north-easterly winds during the winter months to southerly winds during the summer months. Due to the strong solar irradiance in summer a through is formed above the continent and warm and moist air masses are transported from the South China Sea onto the continent. The mean annual precipitation is 1720 mm. In the rainy season, which is lasting from April to September, monthly mean precipitation rates range from 100 mm to 300 mm per month. During wintertime the solar irradiance is low and cools down the continent. This results in a continental anticyclone that transports dry air masses southward. During the winter month, the monthly mean precipitation lies between 30 mm and 90 mm (China Meteorological Administration, 2012).

The rainfall pattern during the winter and spring season 2011/2012 was different. After initially strong rainfall in November 2011 dry weather prevailed, so that no precipitation was registered in December 2011. From January 2012 to mid March 2012 there were numerous days with precipitation resulting in 105 mm in January, 74 mm in February and 66 mm in March 2012, (National Climatic Data Center, 2012). This was caused by a less pronounced continental high pressure so that not dry air masses from the continent but moist air masses arriving from the sea were dominant. The increased precipitation activity in South-East Asia in 2012 is also attributed to the strong "La Niña" event in 2010-2012 (e.g., Boening, 2012). In April 2012 the monsoon season started in Guangzhou with more than 1000 mm precipitation until June 2012. The climatological mean value for April to June is around 700  mm.





## 3 Temporal and vertical aerosol distribution

In this section we present several aspects of the aerosol observations: First, an overview of the temporal development of the aerosol distribution over the entire measurement period is given by the AOD measured by lidar and sun photometer. A case study of the highly polluted atmosphere using the lidar profiles shows the typical aerosol distribution with lofted aerosol layers

that contain a considerable amount of the total aerosol. Finally, seasonal mean profiles are calculated from all particle extinction profiles to identify seasonal patters in the vertical aerosol distribution.

### 3.1 Total AOD

An overview over the aerosol conditions during the entire observation period is given by the derived AOD values. Fig. 1 shows the AERONET (Holben et al., 1998) level 2 AOD values at 500 nm measured by the sun photometer and the AOD values

derived from Polly$^{XT}$ Raman extinction profiles at 532 nm. The combination of the sun photometer working only at daytime and the Raman capabilities for lidar available during night-time offers the unique possibility to obtain a continuous time series of AOD whenever atmospheric conditions allow for it. The extrapolation procedure to calculate the AOD from the lidar profiles seem to underestimate the amount of aerosol close to the ground since it leads to slightly lower AOD values from the lidar than the ones measured by the sun photometer. On the other hand the sun photometer AOD measurements by may be influenced by

more humidity during daytime.

During November and December 2011, sun photometer AOD values are available on 17 and 22 days, respectively. After this period a lot of rain and cloudy weather was present over Guangzhou. As a consequence, for 2012, level 2 sun photometer AOD are only available on 28 days in total. However, these data still provide an overview over the development of the aerosol content throughout the observation period. In the beginning of the observations, in November and December 2011, most AOD

values range between 0.2 and 0.6 with some peak AOD periods with high values up to 1.4. The monthly mean AOD in November was moderate with values of AOD=0.45 measured by the sun photometer and AOD = 0.49 measured by the Lidar. In December these values were AOD=0.49 / 0.39 (sun photometer / lidar). In January (AOD=0.57 / 0.24) and February 2012 (AOD=0.93 / 0.67) only a few observations were available to calculate the mean values, which were higher than in November and December. Unusual heavy rainfall was observed which seems to be triggered by the strong La Niña event during that

winter. In March 2012, a period with very high AOD values started to evolve. The monthly mean AOD of 1.16 / 0.84 was the highest value reached during this field campaign. This is plausible, since no more precipitation was observed that could remove the particles from the atmosphere.

The maximum AOD of 1.95 was measured by the sun photometer on March 28, 2012, which is a very high value compared to the rest of the time series. This high AOD may be the result of hygroscopic growth at the top of the boundary layer, which

was observed by the lidar at the time of this measurement and may not be identified by the sun photometer. Towards the summer season with the onset of the monsoon, the mean AOD decreases to 0.82 / 0.60 in April and 0.49 / 0.34 in May. In the beginning of June the mean AOD value was 0.32 / 0.36. For the whole observation period, the mean photometer AOD value at 500 nm was 0.54 ± 0.33 and the lidar AOD value at 532 nm was 0.47 ± 0.32. This is a little higher than the mean AOD





derived from the only other long term lidar observations in Guangzhou. Hara et al. (2011) measured an annual mean AOD of 0.41 at 532 nm and observed seasonal AOD variation with peaks in springtime and autumn. These mean AOD values indicate a generally high mean background level of aerosol in the atmosphere above the PRD. For comparison, in Leipzig, Germany, a continental Central European site, the yearly mean AOD measured by sun photometer during recent years lies between 0.15 and 0.19 at 500 nm.

## 3.2 Case study

Fig. 2 shows the temporal evolution of the attenuated backscatter coefficient (calibrated range-corrected signal) at 1064 nm for a five day period from March 23 to 29, 2012 in the upper panel. The colors range from low backscatter signal in blue and to high backscatter signal in red. White colors that can be seen on top of the aerosol layers indicate mostly clouds. The blue vertical stripes occur if no signal is measured at all, as for example during a precipitation event on Mach 23 starting around 16:00 UTC. Several precipitation periods follow until about 10:00 UTC the next day. This plot shows the evolution of the aerosol layers that persist over these days. The planetary boundary layer (PBL) is developing every day starting around 00:00 UTC (08:00 LT) and is reaching up to heights of about 2 km - 2.5 km at its maximum. Above the PBL, another aerosol layer is visible, which is reaching from 2 km to 5.5 km in the beginning of the period and to 5 km during the following two days. The large heights of the lofted layers, as observed in this case, were mostly reached in spring. In the lower panel of Fig. 2 the volume depolarisation ratio at 532 nm is shown. Green and yellow colors show considerable volume depolarisation ratios. A narrow layer of elevated depolarisation ratio is visible just above 2 km height. This layer corresponds to the lower boundary of the observed lofted layer. These are most probably large, non-spherical particles which sediment out from this layer.

Lidar profiles of the derived optical properties for March 26, 2012 from 18:00 to 20:30 UTC are presented in Fig. 3. The profiles of the particle backscatter coefficient, the particle extinction coefficient, and the resulting lidar ratio, the backscatter related Ångström exponent, and the linear particle depolarization ratio are shown. Local time of this measurement is 02:00 - 04:30 h. The nocturnal PBL shows a two layer structure with higher particle backscatter and extinction coefficients below 1 km towards the ground and lower values between 1 and 2 km height. From 2 to 5 km height the pronounced lofted aerosol layer is visible with high values of the particle extinction coefficient of up to 300 $Mm^{-1}$ at 532 nm. The mean lidar ratio of the lofted layer is 45.8 sr $\pm$ 7.5 sr at 355 nm and 51.7 sr $\pm$ 8.3 sr at 532 nm. The Ångström exponent shows values around 1.5 throughout the whole profile. The particle depolarization ratio is about 9 % below 2 km height and increases to 15 % below the lofted aerosol layer before it decreases to less than 5 % inside the lofted layer.

Fig. 4 shows HYSPLIT 48 h backward trajectories calculated for arrival heights of 500 m, 2200 m, and 3000 m. The heights represent the observed layer structure, where 500 m is inside the lower PBL, 2200 m is the lower part of the lofted layer, where the depolarisation ratio is elevated, and 3000 m is at the maximum of the particle backscatter and extinction coefficient values inside the lofted layer.

All trajectories remained quite close to the measurement site for the last 48 h. The trajectory at 500 m was coming from the South Chinese Sea bringing some air with lower pollution levels. The trajectory arriving at 2200 m height stayed closest to the measurement site. It came from the north and circled above the measurement site at the same height for one day. The trajectory





arriving at 3000 m is representing the lofted layer. It was rising from ground level just the day before and came from local and regional sources north-west of the measurement site.

This observed structure of particle layers frequently occurred above the measurements site: A PBL with a depth of 1.5 to about 2 km and a decoupled, lofted aerosol layer above. The lofted layers in this case study had a depth of 2.5 to 3 km, which
was the highest depth observed. Typically these lofted layers had a depth between of 1 km and 2 km. The top boundaries of the lofted layers lied between 1.5 km and 5.5 km, as in this example. The optical properties of these lofted aerosol layers will be discussed in more detail in section 4.

### 3.3 Mean profiles

An overview over all evaluated lidar profiles of the particle extinction coefficient at 532 nm are plotted in Fig. 5. In total, 99
single profiles (plotted in grey) were considered for this analysis. To identify seasonal variations in the profile shape, four mean profiles over two month each were calculated and are plotted in bold, colored lines. Note that the scale of the x-axis is as high as 1000 $\text{Mm}^{-1}$ in this plot.

The mean November-December profile was calculated from 35 single profiles. It shows a smooth decrease of the particle extinction coefficient with height and reached zero values at about 4 km. The highest value in the boundary layer was 275 $\text{Mm}^{-1}$,
at 300 m which is the lowermost height of the profiles.

The mean January-February profile is resulting from 11 single profiles only, which is due to the unusual rainy season that winter. The mean profile shows two minima at 1.5 and 2 km height, and a weak mean lofted layer between 2 and 4 km. Further inspection of the single profiles show, that three cases with pronounced lofted layers in the end of February are included here. Also the mean January-February profile reached zero values at about 4 km height. The largest PBL value of the mean particle
extinction coefficient was as high as 500 $\text{Mm}^{-1}$, which is the highest seasonal mean value.

The mean March-April profile calculated from 20 single profiles is the most outstanding. The decrease in the PBL is less pronounced and a clear lofted layer is present between 2 and 4.5 km height with particle extinction coefficients of up to 170 $\text{Mm}^{-1}$ at at 2.9 km height. This is more than four times of the values for the three other mean profiles. In total the mean March-April profile reached zero values at 5.5 km, the largest layer top height out of the four seasonal mean profiles. The
largest mean particle extinction coefficient value in the PBL was 330 $\text{Mm}^{-1}$.

The mean May-June profile was calculated from 33 single profiles and shows increased particle extinction coefficient values between 1 and 2.5 km height. Here, the lofted layers were located at lower altitudes than in the mean spring profile. The amount of aerosol was less as well. The maximum value of the mean particle extinction coefficient in the PBL was 310 $\text{Mm}^{-1}$.

In total, the mean particle extinction coefficient profiles show that particles are present in lofted layers that reach up to heights
of around 5 km. During spring these lofted layers even reached up to a mean value of 5.5 km. In the following discussion the focus is laid on these lofted layers to identify the aerosol types they contain.





## 4   Lofted aerosol layers

The top heights of the lofted aerosol layers range from a few cases of low top heights of 1.5 km up to heights of 5 km (Fig. 6). The highest frequency of occurrences was observed from 2 km to 3 km, and a second smaller peak at 4 km and 4.5 km. During the winter season (Nov 2011 - mid Feb 2012), mostly the low top heights of the lofted layers below 2.5 km were observed and only a few cases with higher top heights occurred. The highest aerosol layer tops were observed during the spring season (end of February, March and April). In total, 21 cases of lofted layers with top heights of 3.5 km and higher occurred. During the summer months May and June the top heights were always between 2.5 km and 3 km height. In total, a variability of the top heights of lofted layers from 1.5 to 5 km was observed, with a majority of 79 cases between 2 and 3 km.

The depths of the observed lofted layers range from a few cases with less than 0.5 km up to cases of 3 km depths. The maximum frequency of occurrence with 74 cases lied between 0.5 and 1.5 km, and in 23 cases the depth of the lofted layer was between 1.5 and 3 km.

### 4.1   Lofted layer AOD

The lidar profiles of the case study shown in Fig. 3 is an example with very high aerosol content. The AOD on this day was among the highest values during the entire observation period: 1.04 at 355 nm and 0.60 at 532 nm. The AOD inside the lofted layer was 0.76 at 355 nm and 0.47 at 532 nm. The AOD-ratio, which is the ratio of the AOD inside the lofted layer to the total AOD, is 73% at 355 nm and 78% at 532 nm in this case. This is partly due to the low AOD inside the lower layers on this day, since the PBL is rather clean at this time of the night. But also two days before, on March 24 and 25 (see Fig. 2), when the PBL was not so clean, the AOD-ratio for the lofted layers was about 70%.

The mean AOD for the lofted layers for each month and some seasonal periods are presented in Table 1. The mean AOD-ratio for the lofted layer in March was the highest with 56%. Also April and June show high AOD-ratios with 48% and 44%, respectively, while the AOD-ratio for May is only 26%. During the winter months November, December, and January, the AOD-ratio was between 12% and 18% while in February it was with 22% already a bit higher. This is also due to three observations with higher aerosol content in the lofted layers in the end of the month. When counting these profiles rather for the spring period, the winter mean AOD-ratio was 15% and the spring mean AOD-ratio value was 48%. For the two summer months, the mean AOD-ratio was 34%. For the whole observation period, the mean AOD-ratio for the lofted layers was 32%.

Thus, a significant part of the aerosol over the PRD is present at high altitudes - especially during spring and summer. Obviously the aerosol can remain in these upper layers for some days, before it may be washed out by rain or is diluted by transport processes.

### 4.2   Aerosol classification

The lidar optical properties used to characterize the aerosol type of the observed particles are the the lidar ratio, the linear particle depolarization ratio, and the Ångström exponents. In Fig. 7, the mean values of the lidar ratio at 532 nm is plotted versus the linear particle depolarization ratio at 532 nm (a) and the backscatter-related Ångström exponent at 355 nm / 532 nm





(b) for all observed lofted layers. For each data point the lidar measurements were lasting over at least 2 hours. Each month has been coded by a color for easier identification. Fig. 7 (a) shows that most data points range between lidar ratios of 30 sr and 80 sr, while the linear depolarization ratio lies mainly below 10 %, and for the majority of cases even below 5 %. This means that most of the time spherical particles were observed. Depolarisation ratios below 5 % and high lidar ratios up to

80 sr indicate freshly produced smoke and pollution arising most likely from local sources. This is true for the observations during winter and also for May and June. In March and April the depolarisation ratio lies more often between 5 % and 10 % and the corresponding lidar ratios are between 40 sr and 60 sr. These properties fit to more aged particles and may be a result of the longer residence time of the particles inside the lofted layers during this time of year. They may also indicate traces of dust (e.g. agricultural dust, road dust, or dust from biomass burning fires injected into the atmosphere) or other large, non-

spherical particles (e.g. dried marine particles) as observed in the case study. Only a few cases of high linear depolarization ratios >10 % were observed. The highest values of more than 20 % were measured on December 2 and 3, 2011 during the only dust advection event that took place during the measurement period. This event, where the dust did arrive from the desert areas north of China, was discussed in detail in Heese et al. (2012).

    Further classification can be done by using the size dependent information given through the Ångström exponents. In

Fig. 7 (b) the backscatter related Ångström exponent for 355 nm/532 nm is shown exemplified. The two other Ångström exponents show a comparable behaviour. It can be seen that the majority of Ångström exponents lies between 1 and 1.5 at lidar ratios between 40 and 60 sr. These properties indicate particles from urban pollution of small and medium size. Pollution mixed with larger particles can be identified by lidar ratios around 50 sr and Ångström exponents down to 0.5. Lidar ratios below 40 sr indicate that marine particles may be mixed with the pollution particles. Pure marine particles are larger and would

have a lower Ångström exponent of 0.1 to 0.3 (Müller et al. , 2007). Another group of observed high Ångström exponents from 1.3 to 2 with lidar ratios from 60 to 80 sr and low depolarisation ratios are identified as more absorbing and smaller burning products that contain soot particles.

    These results are consistent with finding from former studies. Tesche et al. (2007) measured high levels of aerosol pollution over the PRD in October 2006 with lidar ratios between 40 and 55 sr and mean value of $47 \pm 6$ sr. (Ansmann et al. , 2005)

concluded that these lidar ratios are consistent with the presence of large, absorbing particles.

    Baars et al. (2016) have evaluated the derived optical properties for all available PollyNET measurements performed with several Polly instruments at different sites worldwide until 2014 (for more details see also www.polly.tropos.de). From their study they found linear depolarisation ratios at 355 nm and 532 nm below 5% and corresponding lidar ratios between 30 sr and 80 sr for urban particles and burning products.

Groß et al. (2015) report a mean linear depolarisation ratio of $6 \pm 1$ % and a mean lidar ratio $56 \pm 6$ % for anthropogenic pollution in their aerosol classification scheme at 532 nm derived from several field campaigns.

    A summary of lidar optical properties at 355 nm that will be used for the particle classification scheme for the upcoming EarthCARE satellite are presented in Illingworth et al. (2015). They also report linear depolarisation ratios for smoke plumes and anthropogenic pollution below 5 % and lidar ratios from 30 sr to 80 sr and from 45 sr to 65 sr, respectively.



A study by Cattrall et al. (2005) is using AERONET sites for aerosol classification and found that particle optical properties over South-East Asia are distinct from those over other urban/industrial centers, owing to a greater number of large particles relative to fine particles.

In summary, we can conclude that particles of urban pollution arising from traffic, combustion of fuel, industrial and other burning processes are dominant over the PRD. Dust advection only plays a minor role, which was already observed by the Asian dust lidar network (e.g. Nishizawa et al. (2010)).

### 4.3 Origin of the aerosol layers - trajectory analysis

A backward trajectory analysis was performed using the HYSPLIT model (Stein et al., 2015) to determine the origin and the sources of the observed aerosol layers. Backward trajectories were calculated for 144 h for the periods of all evaluated lidar profiles. In most cases three arrival-heights were sufficient to cover the vertical aerosol structure, but in a few cases in which the aerosol extend up to very high altitudes, up to 5 arrival-heights were necessary. Arrival heights were set to the middle of an existing aerosol layer. From this analysis, a total number of 413 backward trajectories was obtained and a cluster analysis was performed to identify the main regions of the particle sources. For the cluster analysis all backward trajectories were divided into arrival heights below and above 1200 m to separate the PBL and the lofted aerosol layers. Additionally, the lofted layers observed at very high altitudes above 3500 m were analysed separately since they show a seasonal dependence and occurred mainly in spring. In the following the results for these three layer categories are presented.

**Lower layers - PBL**

For this category the lowermost layers with a top height reaching up to a maximum of 1200 m were considered. This covers mainly the PBL and only a few cases of very low lofted layers may be included. The number of clusters that were calculated was 5 (Fig. 8 a). Most of the trajectories arriving at these layers came from local and regional sources. 37 % of these local/regional trajectories came from north-easterly directions (cluster 1), 29 % from south-westerly directions (cluster 2), and 15 % came from westerly directions (cluster 3). Only about a fifth of all trajectories come from regions further away: 10 % of the trajectories come from westerly directions (cluster 4), reaching back to the Arabian and Saharan deserts and 9 % of the trajectories (cluster 5) come from north-westerly directions, originating from the areas north of China. This is a results that was expected for the lower altitudes above the PRD. Most air masses are of local origin, either from the mainland or from the close sea.

**Lofted layers**

The air masses arriving at the layers above 1200 m mainly came from the westerly and north-westerly directions (Fig. 8 b). 23 % of the air masses come from north-westerly regions (cluster 2) and 31 % came from closer sources from the West (cluster 3). Only 7 % of the trajectories indicate a possible source from farther arriving from westerly directions (cluster 5) and only 2 %





of the trajectories arrive from far sources from North-west (cluster 4). Also for this category we can conclude that the majority of the aerosol burden originated from local sources that are close to or inside the PRD region.

**Highest layers**

A separate cluster analysis was performed for the highest observed layers above 3500 m. Here, only three clusters were

calculated. However, the behaviour of these trajectories is comparable to lofted layer trajectories. 35 % of the trajectories came from westerly directions (cluster 3), 28 % of the trajectories came from slightly more north-westerly directions (cluster 2), and, again, 37 % of the trajectories came from local regions (cluster 1, Fig. 8 c). Thus, no significant difference concerning the origin was found for the particle layers observed up to very high altitudes in March and April 2012 compared to the lower lofted layers.

In summary we can conclude that only a very low percentage of aerosol is transported to the PRD from sources further away. About one third of all trajectories show air mass transport from close by, even if they were calculate for 6 days.

## 4.4 Statistics of lofted layer optical properties

In order to give a comprehensive overview over the typical aerosol conditions in the PRD, a statistical analysis of the measured optical properties of the observed lofted aerosol layers is given in the following. Statistical results are of importance if the data shall be used for further analyses, for example for modelling studies. In Fig. 9, the mean and median values and the standard

deviations were calculated for the following properties: The lidar ratios at 355 nm (a) and 532 nm (b), the linear particle depolarisation ratio at 532 nm and the Ångström exponents related to the extinction coefficient between 355 nm and 532 nm (d), the particle backscatter coefficient between 355 nm and 532 nm (e), and particle backscatter coefficient between 532 nm and 1064 nm (f).

The statistics of the lidar ratios show a wide range of values from 30 to 80 sr for both wavelengths (Fig. 9 (a) and (b)). Most of the lidar ratios lie between 30 and 60 sr, with a peak value of 50-55 sr at 355 nm and 40-50 sr at 532 nm. The mean lidar ratio at 355 nm is 50.7 and slightly higher than the mean lidar ratio at 532 nm of 48.1 sr . These values confirm the rare and very short former measurements done by lidar in the PRD region. From the first Megacity campaign in November 2009 (Chen at al. , 2014) only lidar ratios at 355 nm were measured. They were $64 \pm 10$ sr for a hazy period and $56 \pm 9$ sr for a

moderate polluted period. Lidar ratios at 532 nm were measured in Xinken (PRD) in October 2004, and the mean value of the lidar ratio was $46.7 \pm 5.6$ sr (Ansmann et al. , 2005; Tesche et al. , 2007).

As already seen in the aerosol classification (Fig. 9 (c)), the statistics of linear particle depolarisation ratio at 532 nm show that for the majority of 89 cases the depolarisation ratio was below 5 % and only 11 cases lie between 5 and 10 %. In only 6 cases the depolarisation ratio was higher than 10 %. This implies that mostly spherical particles were present and that non-

spherical particles are seldom mixed into the aerosol. The mean value for the linear depolarisation ratio is 3.6 % $\pm$ 3.7 %.

The histogram of the Ångström exponent related to the particle extinction coefficient at 355 nm to 532 nm (Fig. 9 (d)) show a few cases of low values below 1, a peak of 45 cases between 1.0 and 1.5, and a smaller peak of 26 cases from 1.5 to 2. Even





higher values up to 3.0 were observed in 14 cases. The mean value for the extinction related Ångström exponent is $1.48 \pm 0.49$.

The Ångström exponents related to the particle backscatter coefficients at 355 nm / 532 nm (Fig. 9 (e)) and 532 nm / 1064 nm (Fig. 9 (f)) show a comparable distribution. Most cases, 51 and 49, respectively, are lying between 1. and 1.5 and 27 and 24 cases, respectively, between 1.5 and 2.0. Only two cases lie above 2.0. Below 1.0, however, the number of cases is slightly higher for the particle backscatter related Ångström exponent for 532 nm / 1064 nm, and especially the number of cases below 0.5. is higher. The mean values for the backscatter related Ångström exponent are $1.28 \pm 042$ for 355 / 532 nm and $1.17 \pm 0.49$ for 532 / 1064 nm. Only a few cases show Ångström exponents below 0.5 that indicates larger particles in the coarse mode. This is consistent with the few observed cases of linear particle depolarisation ratios above 10 %. The derived mean particle optical properties are summarized in Table 2.

## 5  Conclusions

For the first time, continuous multi-wavelengths Raman and polarization lidar observations were performed over a long-term period in the highly polluted atmosphere of the Pearl River Delta, Guangzhou, China. The measurements were taken from November 2011 to mid June 1012. The observations show that a high load of aerosol is present not only in the planetary boundary layer but also in lofted layers that reach up to several kilometers height. The heights of these lofted layers show a seasonal dependence with heights below 2 km during winter up to very high layer top heights above 5 km in spring. The aerosol optical depth in the lofted layers make a significant part of the total AOD observed in the vertical profile. The total AOD is rising from monthly mean winter values of 0.49 in November to 0.84 in March. The percentage of the AOD in the lofted layers varies from 12 to 20 % in the winter months to 48 and 55 % in spring.

These results confirm a previous study by Wang et al. (2011) who investigated seasonal variations of AOD over different locations in China using the Chinese Sun Hazemeter Network. In most parts of China, AODs are at a maximum in spring or summer and at a minimum in autumn or winter. This was also observed in Guangzhou and is consistent with the Asian monsoon circulation in the region.

A classification of the observed aerosol in the lofted layers over the PRD by using the lidar optical properties shows mostly low linear particle depolarization ratio and the wide range of the lidar ratios. These properties indicate that mainly particle mixtures of urban pollution arising from traffic, combustion of fuel, industrial and other burning processes are present in these layers. The particles are locally and regionally produced and are only seldom mixed with transported particles from further away. During the summer monsoon season they may also be mixed with particles of marine origin from the close-by sea. Dust mixtures into the pollution aerosol was only observed in one case.

*Acknowledgements.* The measurements were conducted in the frame of the special priority program "Megacities-Megachallenge - Informal Dynamics of Global Change" (SPP 1233) funded by the German Research Foundation (DFG).

We thank our colleagues from the Sun Yat-sen University, Guangzhou, for their support throughout the measurement campaign.





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





**Table 1.** Monthly mean values of total AOD and AOD of identified lofted layers derived from lidar extinction profiles.

| Period | total AOD | Layer AOD-LL | Percentage |
|---|---|---|---|
| November 2011 | 0.49 ± 0.29 | 0.06 ± 0.05 | 12.2% |
| December 2011 | 0.39 ± 0.26 | 0.07 ± 0.09 | 18.0% |
| January 2012 | 0.24 ± 0.07 | 0.03 ± 0.02 | 12.5% |
| February 2012 | 0.67 ± 0.46 | 0.15 ± 0.11 | 22.4% |
| March 2012 | 0.84 ± 0.27 | 0.47 ± 0.19 | 55.9% |
| April 2012 | 0.60 ± 0.41 | 0.29 ± 0.29 | 48.3% |
| May 2012 | 0.34 ± 0.26 | 0.09 ± 0.12 | 26.5% |
| June 2012 | 0.36 ± 0.21 | 0.16 ± 0.16 | 44.4% |
| November 2011 - February 2012 | 0.40 ± 0.27 | 0.06 ± 0.07 | 15.0% |
| February 2012 - April 2012 | 0.77 ± 0.34 | 0.37 ± 0.24 | 48.2% |
| May 2012 - June 2012 | 0.35 ± 0.24 | 0.12 ± 0.14 | 34.3% |
| November 2011 - June 2012 | 0.47 ± 0.32 | 0.15 ± 0.19 | 32.0% |

**Table 2.** Summary of lidar mean optical properties for the lofted aerosol layers measured over the PRD region.

| property | mean ± std |
|---|---|
| AOD-LL 532 nm | 0.15 ± 0.19 |
| Lidar Ratio 355 nm (sr) | 50.7 ± 9.1 |
| Lidar Ratio 532 nm (sr) | 48.1 ± 11.0 |
| Ång EXT 355-532 nm | 1.48 ± 0.49 |
| Ång BSC 355-532 nm | 1.28 ± 0.42 |
| Ång BSC 532-1064 nm | 1.17 ± 0.49 |
| Depol Ratio (%) | 3.6 ± 3.7 |

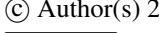



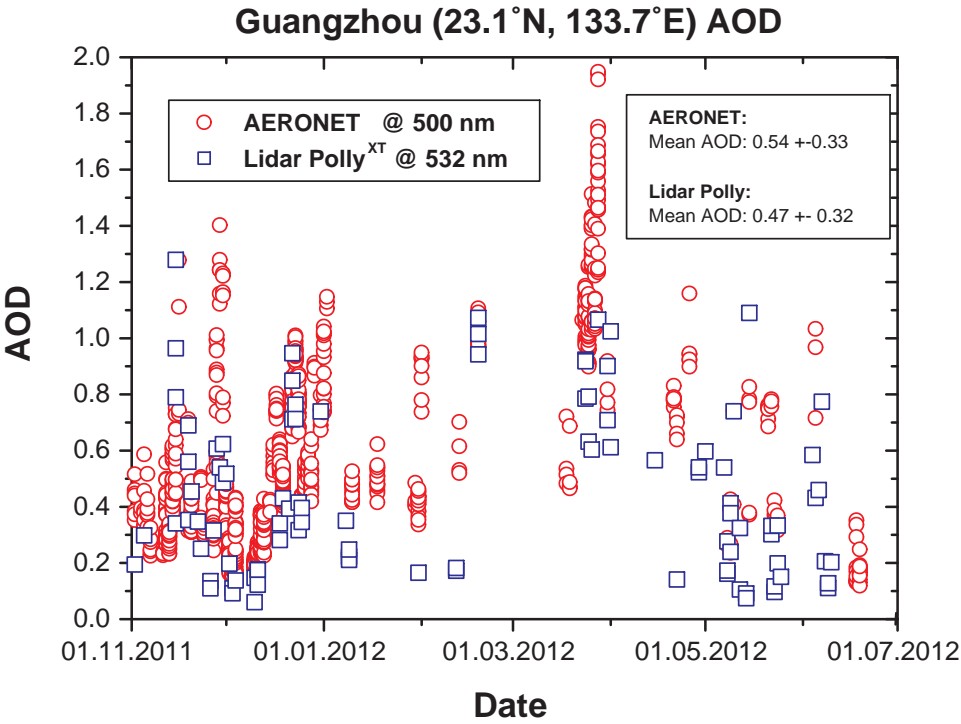

**Figure 1.** AOD derived from Polly$^{XT}$ Raman extinction measurements at 532 nm and AERONET level 2 aerosol optical depth derived from sun photometer measurements at 500 nm.



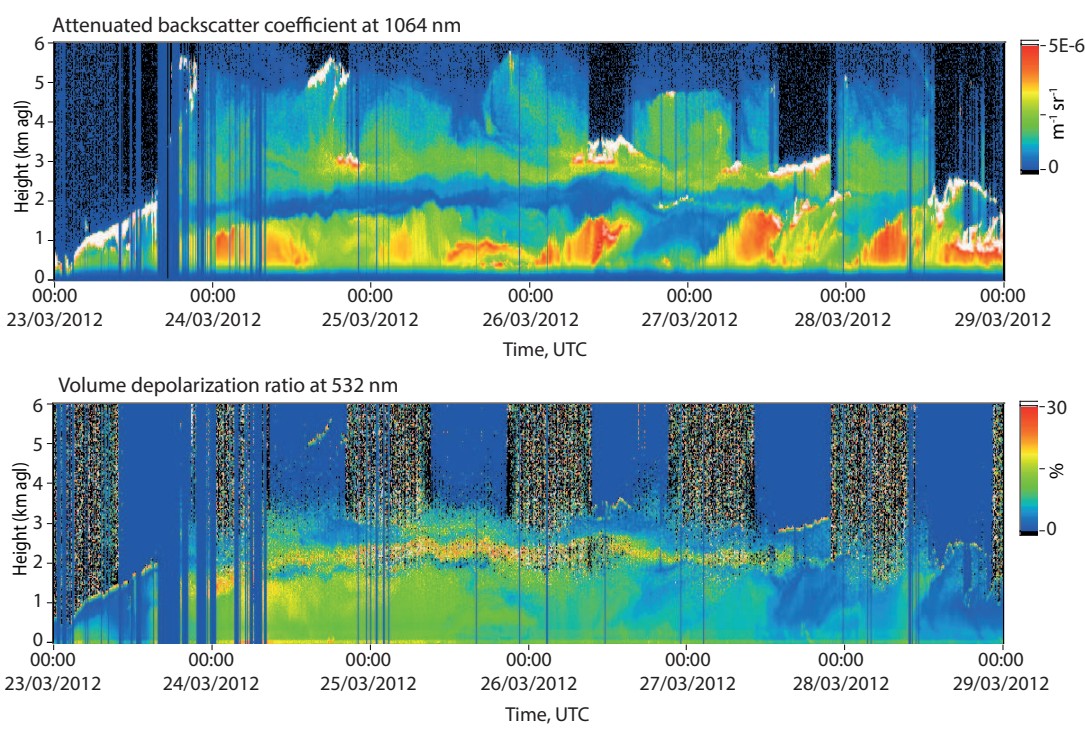

**Figure 2.** Attenuated backscatter coefficient at 1064 nm (upper) and volume depolarization ratio (lower) for the 5-day period from March 23 to March 28, 2012.





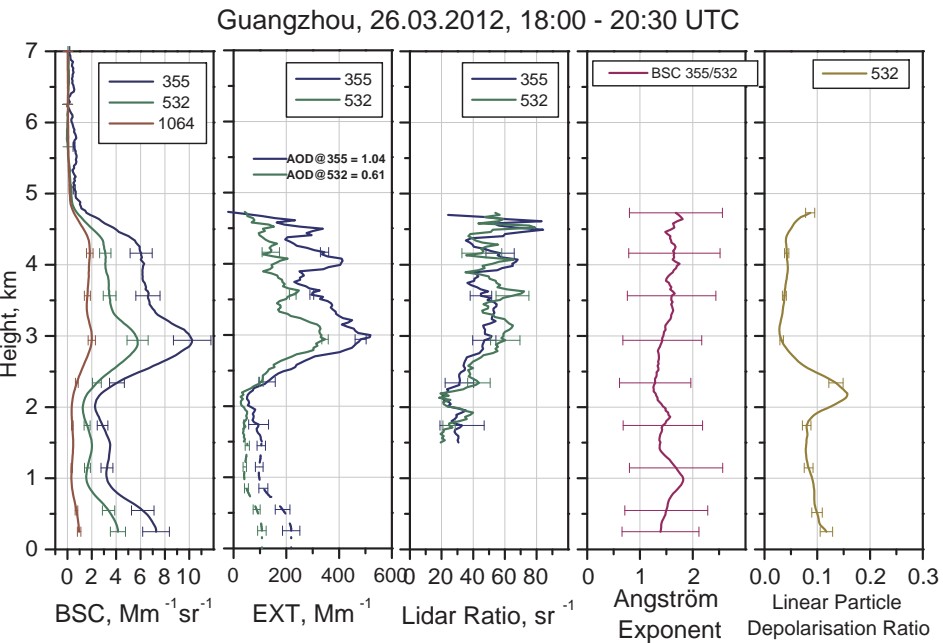

**Figure 3.** Lidar optical property profiles of particle backscatter coefficient (BSC), particle extinction coefficient (EXT), lidar ratio, Ångström exponent and linear particle depolarization ratio at 532 nm derived from Polly$^{XT}$ measurement on March 26, 2012 (local time is + 8 h, 02:00 to 04:30 LT)). The profiles are vertically smoothed over 15 rangebins which corresponds to 450 m. The dashed prolongation of the particle extinction profiles towards the ground result from fitting the BSC to the EXT profile and extrapolation to the ground.



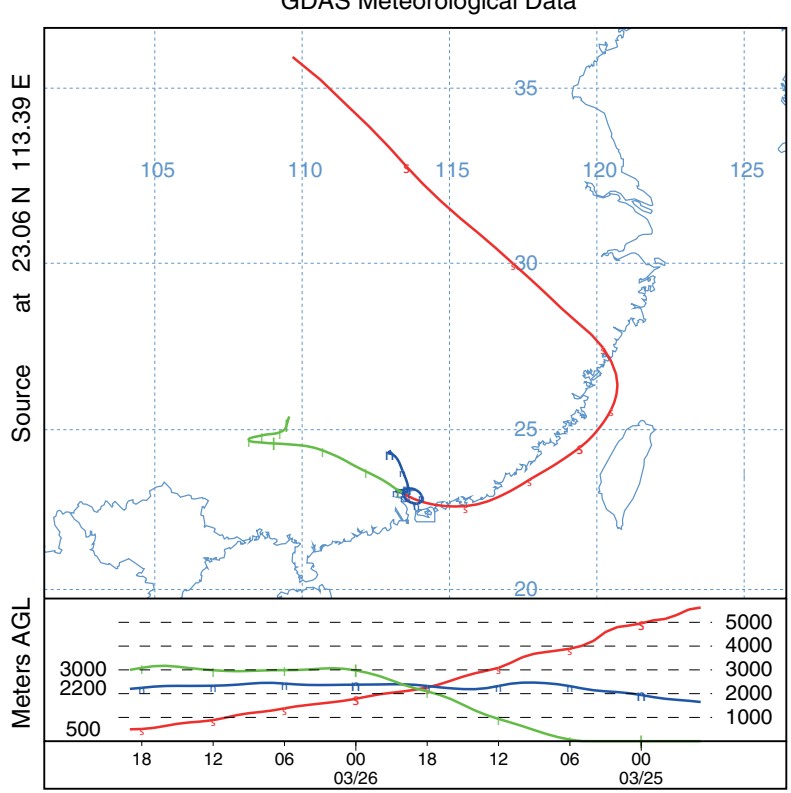

**Figure 4.** Trajectory analysis for March 26, 2012, 48 h back in time, arriving at 500 m inside the boundary layer, at 2200 meter below the lofted layer, and at 3000 m inside the lofted layer.





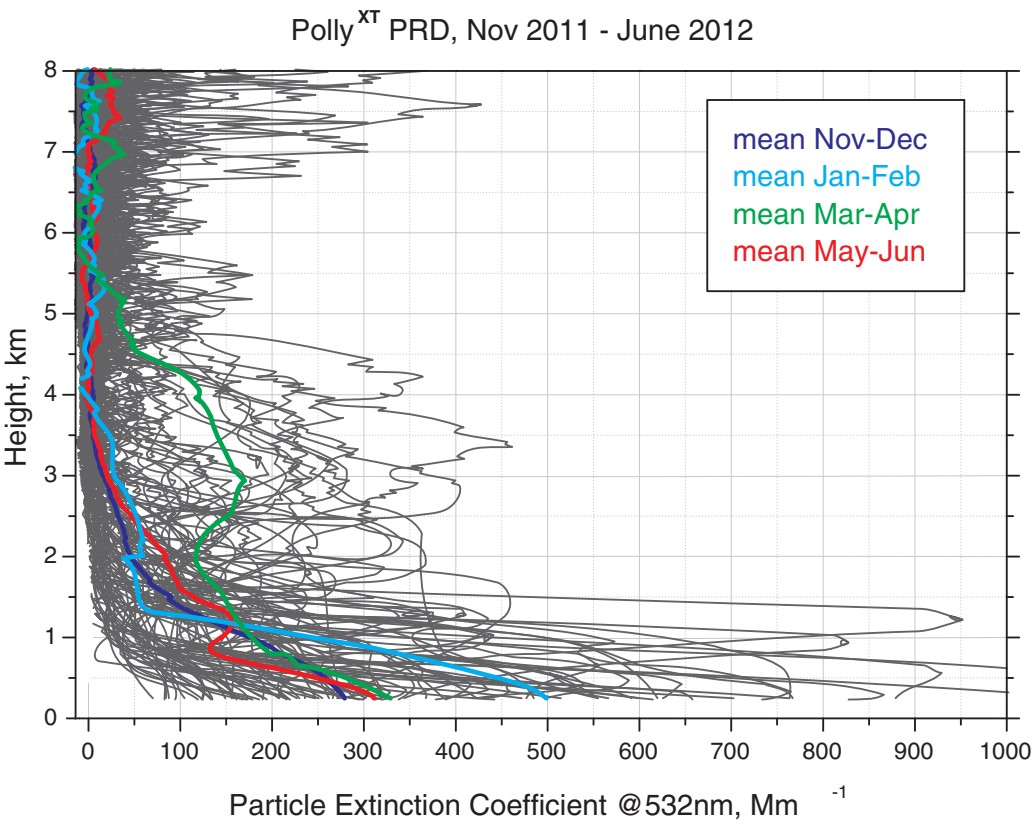

**Figure 5.** All single and seasonal mean particle extinction profiles measured the by the lidars at 532 nm during the entire observation period from November 2011 to June 2012.





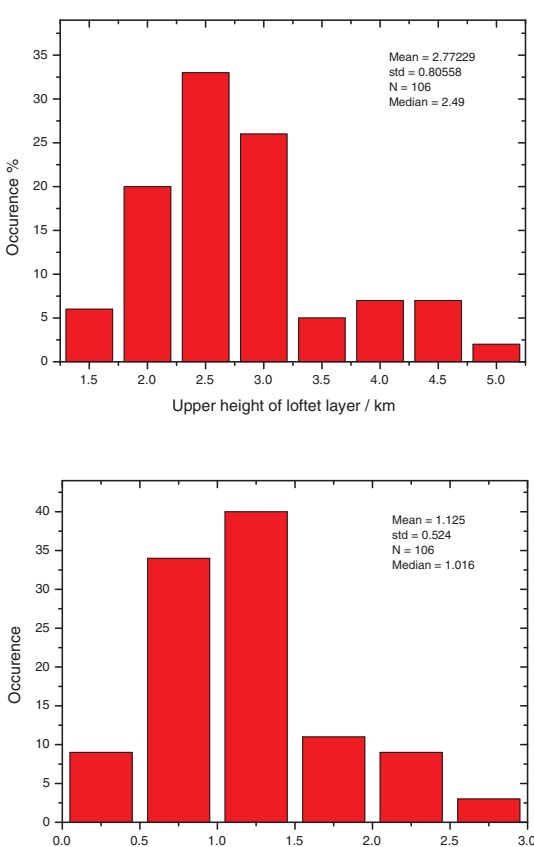

**Figure 6.** Histogram of lofted layer heights and depths





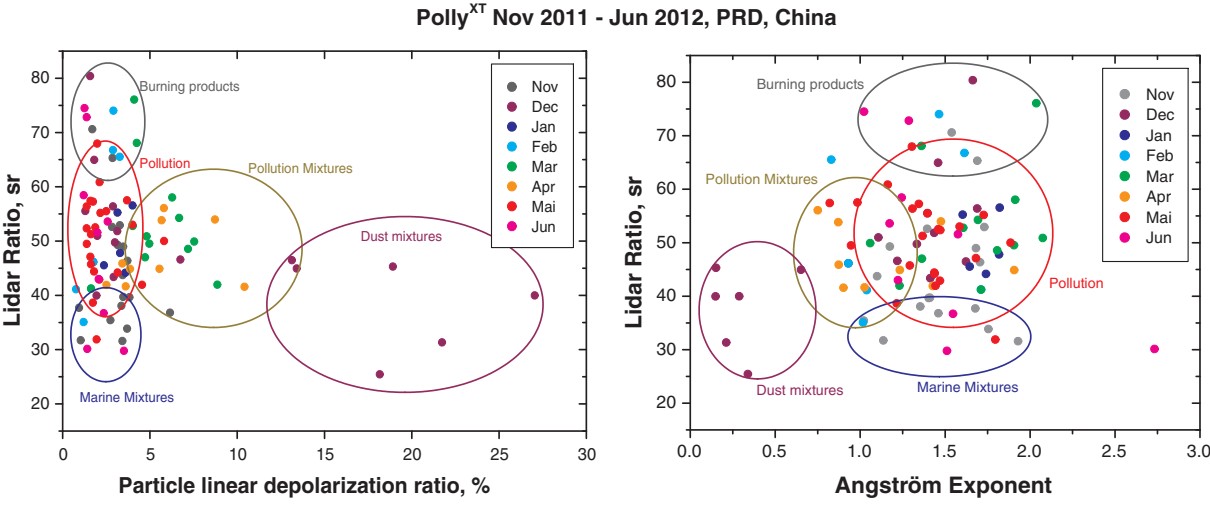

**Figure 7.** Aerosol classification using lidar ratio at 532 nm versus linear particle depolarization (left panel) and Ångström exponent for the particle backscatter coefficient for 532 nm / 1064 nm (right panel) for lofted layers in the period from November 2011 to June 2012. The colored circles indicate the identified particle mixtures and corresponding measurements.





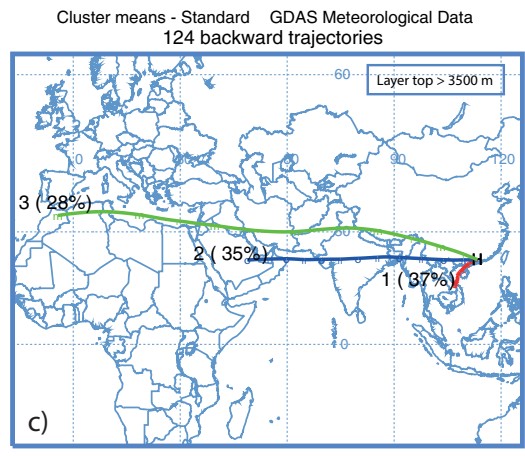

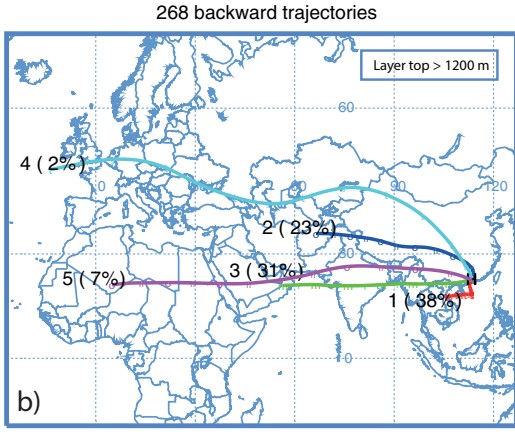

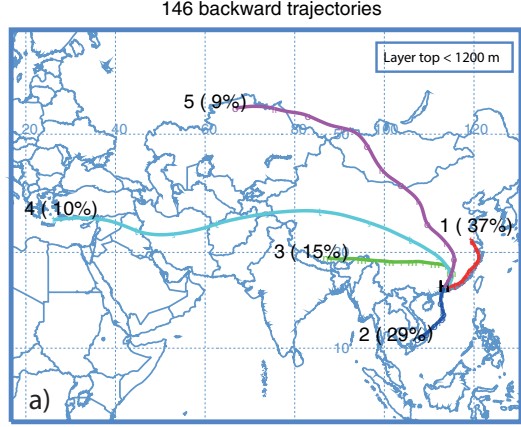

**Figure 8.** Trajectory cluster means for (a) PBL top heights up to 1200 m, (b) all lofted layers above 1200 m, and (c) lofted layers top height above 3500 m. All trajectories were calculated 144 h back in time.



**Figure 9.** Statistical analysis of the optical properties of the observed lofted aerosol layers: (a) Lidar ratio at 355 nm, (b) Lidar ratio at 532 nm,

(c) linear particle depolarisation ratio at 532 nm, (d) Extinction Ångström exponent, (e) Backscatter Ångström exponent (355 nm / 532 nm),

(f) Backscatter Ångström exponent (532 nm / 1064 nm).