# Peer review of "CONTINUOUS VERTICAL AEROSOL PROFILING WITH A MULTI-WAVELENGTH RAMAN POLARIZATION LIDAR OVER THE PEARL RIVER DELTA, CHINA"

_Atmospheric Chemistry and Physics, 2016_

## Referee Comment (RC1) · Anonymous Referee #2 · 4 Nov 2016

General comments:

This manuscript presents a statistical analysis for the aerosol optical properties of the polluted atmosphere over the Pearl River Delta, Guangzhou, China, using a multi-wavelength Raman and depolarization lidar and a sun-photometer during November 2011 to mid- June 2012. Multiple range-resolved optical parameters of aerosols are characterized with the monthly and seasonal average, such as the extinction and backscatter coefficient, lidar ratio and depolarization ratio. In particular, the statistics and type classification for the lofted aerosols are analyzed. Overall, the paper is well organized, but some details on the statistical methodology are missed and the English writing needs to be improved.

Specific comments:

1. Generally, the capability of Raman-channel detecting aerosol extinction profile is quite limited in the daytime due to the sky noise. Some related information are missed in the manuscript as follows. What are the valid altitudes for the Raman-channel derived aerosol extinction profile in the daytime and night-time? What's the range of lidar geometric overlap function (GFF) (where the GFF=1)? How long is the time average for calculating aerosol extinction coefficient? Are all the aerosol extinction profiles in this manuscript derived from the Raman-channel in the night?

2. For the statistical analysis such as the monthly average in the Table-1 and Fig.5 and Fig.7, How many days data for each month?

3. In Fig.2 (upper panel), there are a lot of strips or lines that show very small values the whole profile or from the surface to free troposphere (e.g. at 00:00 24/03/2012). They seem artificial; what reasons cause them? In Fig.2 ((lower panel), the clean layers of 2-km altitude show consistently higher depolarization ratios over the days. They seem not in the lower layers of aerosols, it is difficult to understand them. Did you check the possible distortion or nonlinearity of weak signals at those clean air layers?

4. In Fig.3, the lower lidar-ratios (<40 sr) and higher depolarization ratio (∼15%) at 2-km altitude are doubtful since the Angstrom exponents vary little over the altitude. Why are the aerosol extinction coefficient profiles cut below 1.5 km altitude? When calculating the aerosol backscatter profiles with the Raman and elastic-scattering signals, how do you determine the free aerosol or clean-air layer? What heights are generally used?

5. In Fig.5 or in the Line 9 of Page-7, are the single profiles of extinction the daily or hours averaged? Are they calculated from the Raman-channel in the night only?

6. In the Section 4 Lofted aerosol layers. How do you define a lofted aerosol layer, visually or using a threshold of aerosol extinction against the molecular value? Because of

the temporal-spatial variations of lofted aerosol layer, how do you take the layer height, using hourly or daily averaged profile?

7. In the Section 4.2 Aerosol classification, In Fig.7, are the data points the daily averaged values? Those circles marked for the aerosol types seem arbitrary or not objective based on some thresholds of aerosol optical parameters. What are your methods or any thresholds of aerosol optical properties for classifying these aerosol types? For the given type of aerosols, what is the difference between the "Pollution" and "Pollution mixture" aerosol? "Burning product" is a little confused, "biomass burning"?

8. In the Section 4.3 Origin of the aerosol layers- trajectory analysis, The lofted aerosols below 1200-m are probably from the local nocturnal residual layer since they are so low or in the PBL, thus they are probably not from the long-range transport.

Page-10, Line-12, a total number of 413 backward trajectories was obtained. It seems that they are not the daily averaged profiles since your total observation days are less than this number. How long is the time average for a lidar profile? That means that on some day you might have a lot of aerosol profiles while on other days you might only have one or none.

9. In the Section 4.4, If possible, the statistics of PBL aerosols optical properties can be given for the comparisons with the aloft aerosols because the PBL aerosol pollutants are more related to the human health or draw more attentions.

10. In the Section 5 Conclusion, Page-12, Line 22-23, authors mention "This was also observed in Guangzhou and is consistent with the Asian monsoon circulation in the region." There are no enough discussions about the effects of Asian monsoon circulation on the aerosols. how does the Asian monsoon affect the aerosols?

Page-12, Line 26-27, "The particles are locally and regionally produced and are only seldom mixed with transported particles from further away." This is not consistent with the Figure 8 (b) and (c), even Figure 8 (a). For instance, in Fig.8(b), the cluster-3 for

the long-distance transport shows 31% percentage against the 38% of the Cluster-1.

Technical corrections:

1. Page-1, Line-6, two "observed" appear in the sentence. Please delete the first one and move "by the sunphtometer" afterward to the second "observed". Please give the wavelength for the aerosol optical depth and lidar-ratio.

2. Page-1, Line-8, please delete the word "even".

3. Page-1, Line-9, "aerosol" should be "aerosol types".

4. Page-1, Line-11, please add "%" behind the number "3.7".

5. Page-1, Line-12, you may say the mixture of find and coarse-mode aerosols.

6. Page-1, Line-13, the word "mainly" should be "main".

7. Page-2, Line-9, please add the word "for" in front of "most of the time in the PRD".

8. Page-3, Line-21, please revise the word "is increasing" with "increases".

9. Page-3, Line-26, please delete the word "also".

10. Page-4, Line-1, this sentence is confused.

11. Page-6, Line-22, "04:30 h" should be "04:30 am".

12. Page-8, Line-2, please revise the sentence or just say:

"The top heights of the lofted aerosol layers range from a few cases of 1.5 km to 5 km (Fig. 6)."

13. Page-8, Line-9, please delete the word "depths" after " 3 km".

14. Page-8, Line-13, the word "is" should be 'are".

15. Page-9, Line-30, the word "6%" should be "6 sr".

16. Page-11, Line-11, the word "calculate" should be "calculated".

17. Page-11, Line-22, please add "sr" behind the number "50.7".

18. Page-12, Line-7, the number "042" should be "0.42".
* * *

---

## Referee Comment (RC2) · Anonymous Referee #1 · 7 Nov 2016

**General comments**

"Continuous Vertical Aerosol Profiling with a Multi-Wavelength Raman Polarization Lidar over the Pearl River Delta, China" by Heese et al. describes general characteristics of aerosol particles observed by a Raman lidar in China. The southern part of China is highly focused in the air pollutant studies and the fact indicated by this paper would aid understanding of the structure and transportation of air pollutants in the area. The contents of the paper is well organized and the method of the data analysis is almost acceptable. Thus the publication in ACP is recommended after minor revision.

**Specific comments**

[Figure]

P1L12, coarse mode is mentioned here, but only small and medium size are mentioned in P9L16 for same angstrom range 1-1.5. Unify the statements.

P3L32, the interval 2-3h was written, but the original time resolution was not indicated.

P6L18, What are 'large, non-spherical particles'? Are they dried sea salt stated in P9L10?

P6L28 and Fig.4, is there any reason to choose 48 hours for backward trajectories here? Do authors just intend to show the air mass stayed there for long time? If trajectories are introduced to infer the origin of particles, 48 hours are insufficient.

P7L9 and Fig.5, does the particle extinction mean extinction of aerosol particles? In the figure we see several peaks around 7-8km. Are they cirrus or aerosol layers?

P8L2, How were the top heights of the lofted aerosol layers determined?

P9L5, Indicate literature to identify local sources using depolarization ratios and lidar ratios.

P9L19, If authors identify lower Angstrom 0.1-0.3 as marine particles, the lower left part in Fig.7b should be marked similarly.

P10L14, 1200 m was determined from Fig.5?

Technical corrections

P9L30, a mean lidar ratio 56+-6 'sr'

P12L7, 1.28+-0'.'42

————————————————————

---

## Author Response (AR1)

**Answer to Referee #1**

*We thank the referee for the careful reading the manuscript and the kind suggestions to improve it. Please find our answers to the specific comments and technical corrections below. Answers are typed in cursive letters and new text is typed in smaller letters.*

Specific comments

P1L12, coarse mode is mentioned here, but only small and medium size are mentioned in P9L16 for same angstrom range 1-1.5. Unify the statements.

- *There was some mismatch, the statement was adapted in the abstract*

The majority of the Angström exponents were observed between 0.5 and 1.5 indicating a mixture of fine- and coarse-mode aerosols.

P3L32, the interval 2-3h was written, but the original time resolution was not indicated.

- *The original time resolution is 30 seconds and the vertical resolution is 7.5 meters. New sentence P3L15:*

The vertical resolution of the raw profiles is 7.5~m and the data were stored with a temporal resolution of 30~seconds.

P6L18, What are 'large, non-spherical particles'? Are they dried sea salt stated in P9L10?
- *Large, non-spherical particles can in principle be dust particles, or sea-salt particles, or even ice particles (the latter is very unlikely here, due to the altitude and corresponding temperature). We just wanted to express, what a higher depolarization ratio implies. However, in this chapter of the manuscript the particle classification is not the subject, so we leave this sentence out here and come back to the particle classification later.*

P6L28 and Fig.4, is there any reason to choose 48 hours for backward trajectories here? Do authors just intend to show the air mass stayed there for long time? If trajectories are introduced to infer the origin of particles, 48 hours are insufficient.

- *Yes, here we wanted to show, that the air masses came from close by during the last 48 hours. The trajectory cluster analysis discussed in chapter 4.3, for which trajectories were calculated for 144 h, showed that only a few percentage of the air masses come from far away sources.*

P7L9 and Fig.5, does the particle extinction mean extinction of aerosol particles? In the figure, we see several peaks around 7- 8km. Are they cirrus or aerosol layers?

- *Particle extinction coefficient at 532 means the extinction of light by the aerosol particles. Figure 5 caption was corrected accordingly:*

Figure 5. All single and seasonal mean particle extinction coefficient profiles measured by the lidar at 532 nm during the entire observation period from November 2011 to June 2012.

- *Yes, the peaks around 7-8 km height are due to cirrus clouds.*

P8L2, How were the top heights of the lofted aerosol layers determined?

- *The top heights of the lofted aerosol layers were identified visually from the backscatter coefficient profiles. The top height was defined were the backscatter coefficients reach the molecular background, the lower boundary of the lofted layer was set to the minimum in the backscatter coefficient profile between the PBL and the lofted layer. We added this point as follows:*

The top heights and the depths of the lofted aerosol layers are shown in Fig.6. Both values were identified visually using the backscatter coefficient profiles. The top height was defined were the backscatter coefficients reach the molecular background and the lower boundary of the lofted layer was set to the minimum in the backscatter coefficient profile between the PBL and the lofted layer.

P9L5, Indicate literature to identify local sources using depolarization ratios and lidar ratios.

- *We did not want to state that the depolarization ratio and lidar ratios identify the source, but the type of the aerosol. Literature that is reporting similar optical properties is used iin the discussion of these results at P9L33-35.*

*Modified sentence:*

Depolarization ratios below 5 % and high lidar ratios up to 80 sr are caused by particles of low reflection and high absorption capabilities. These are most likely freshly produced smoke and pollution particles arising from local sources.

P9L19, If authors identify lower Angstrom 0.1-0.3 as marine particles, the lower left part in Fig.7b should be marked similarly.

- *The data points of low Angström and low lidar ratio are identified as dust and dust mixtures because the origin of the particles was surveyed by individual trajectories. This is the only dust case of 2-3 December 2011 and is discussed in another paper, see P9L11.13.*
  *Since there are no measurement points of pure marine particles, we did not mark this area separately.*

P10L14, 1200 m was determined from Fig.5?

- *The altitude of 1200 m was determined as the mean/median value of all identified PBL layer tops. They are not shown in a separate figure. However, you are right; the total of all extinction profiles in Figure 5 gives further evidence that it is reasonable to choose this height for the mean upper boundary of the PBL top.*

Technical corrections:

P9L30, a mean lidar ratio 56+-6 'sr' - *done*

P12L7, 1.28+-0'.'42 - *done*

**Answer to Referee #2**

*We thank the referee for the careful reading the manuscript and the kind suggestions to improve it. Please find our answers to the specific comments and technical corrections below. Answers are typed in cursive letters and new text is typed in smaller letters.*

Specific comments:

1. Generally, the capability of Raman-channel detecting aerosol extinction profile is quite limited in the daytime due to the sky noise. Some related information are missed in the manuscript as follows. What are the valid altitudes for the Raman-channel derived aerosol extinction profile in the daytime and night-time? What's the range of lidar geometric overlap function (GFF) (where the GFF=1)? How long is the time average for calculating aerosol extinction coefficient? Are all the aerosol extinction profiles in this manuscript derived from the Raman-channel in the night?

- *To emphasize this aspect, we added the following information to page 3 line 20:*

For the determination of the particle backscatter coefficient and particle extinction coefficient during night-time the Raman method (Ansmann, 1992) was applied.
During daytime the Fernald-Klett method (Klett, 1981, Fernald, 1984) was used, but in this study only the Raman derived profiles were taken into account.

- *The valid altitudes for the Raman channel derived aerosol extinction profiles start at 1.5 km height. Below 1.5 km the data are effected by the geometric overlap. The overlap function could not be determined due to very high aerosol load at these altitudes (see page 4, line 3-6). Therefore, the extinction profiles below 1.5 km were extrapolated downwards using the backscatter profiles derived by the Raman method, where the overlap effect is eliminated by the ratio of two channels. To clarify we added this information in the text on page 4 from line 3:*

The lidar data presented here are without any overlap correction. The overlap function could not be calculated due to permanently high aerosol load in the atmosphere over the PRD. However, to be able to calculate the AOD from the lidar profiles, the Raman backscatter profiles were fitted to the Raman extinction profiles at the heights below 1.5 km height. The Raman backscatter profiles are not affected by the incomplete overlap since a ratio of two channels is used in the respective algorithm.

- *The average time to calculate the extinction coefficient profiles was 2 to 3 hours. This is written in the text at page 2 line 32, just before the overlap paragraph.*

2. For the statistical analysis such as the monthly average in the Table-1 and Fig.5 and Fig.7, How many days data for each month?

- *Nov 19, Dec 24, Jan 5, Feb 6, Mar 12, Apr 9, Mai 21, Jun 10, total number is 106. The same profiles were used for the statistics in Fig 6, 7, and 9.*
- *The number of profiles used in Table 1 are now added to the table caption.*

- *The number of single extinction profiles used for Fig 5 is less (99) and is described in chapter 3.3, at page 7:*
    *line 13: The mean November-December profile was calculated from 35 single profiles.*
    *line 16: The mean January-February profile is resulting from 11*
    *line 21: The mean March-April profile calculated from 20 single profiles*
    *line 26: The mean May-June profile was calculated from 33 single profiles*

3. In Fig.2 (upper panel), there are a lot of strips or lines that show very small values the whole profile or from the surface to free troposphere (e.g. at 00:00 24/03/2012). They seem artificial; what reasons cause them?

- *These blue lines indicate "no signal", i.e. the laser is off due to a rain event or caused by insects flying through the rain sensor. This was mentioned in the text (page 6, line 10). However, for convenience, we added a comment to the figure caption as well.*

Fig. 2: Attenuated backscatter coefficient at 1064~nm (upper) and volume depolarization ratio (lower) for the 5-day period from March 23 to March 28, 2012. The blue, vertical lines in the plots occur when the laser is automatically switched off due to rain events. This may also be caused by insects flying through the rain sensor.

In Fig.2 ((lower panel), the clean layers of 2-km altitude show consistently higher depolarization ratios over the days. They seem not in the lower layers of aerosols, it is difficult to understand them. Did you check the possible distortion or nonlinearity of weak signals at those clean air layers?

- *Yes, these high depolarization ratios are observed in the lower part of the upper aerosol layers. The color scale may be misleading here concerning regions with low aerosol backscattering. It can be better seen in the profiles in Figure 3. We are confident that the data evaluation is correct.*

4. In Fig.3, the lower lidar-ratios (<40 sr) and higher depolarization ratio (_15%) at 2-km altitude are doubtful since the Angstrom exponents vary little over the altitude. Why are the aerosol extinction coefficient profiles cut below 1.5 km altitude? When calculating the aerosol backscatter profiles with the Raman and elastic-scattering signals, how do you determine the free aerosol or clean-air layer? What heights are generally used?

- *Below 1.5 km the geometric overlap of the lidar system is uncomplete. See also answer to question 1.*
- *The aerosol free layer for the Raman calculation is visually determined. Generally, heights above the aerosol layers are used where the signal to noise ratio is still high. These heights lie usually around 10 km altitude. This was also the case for the data in Fig 3.*

5. In Fig.5 or in the Line 9 of Page-7, are the single profiles of extinction the daily or hours averaged? Are they calculated from the Raman-channel in the night only?

- *The single particle extinction profiles are calculated from 2-3 hour Raman measurement. See also answer to 1) above.*

6. In the Section 4 Lofted aerosol layers. How do you define a lofted aerosol layer, visually or using a threshold of aerosol extinction against the molecular value? Because of the temporal-spatial variations of lofted aerosol layer, how do you take the layer height, using hourly or daily averaged profile?

- *Also here, 2-3 hour averaged measurements were used for to calculate the profiles. The top heights of the lofted aerosol layers were identified visually from the backscatter coefficient profiles. The top height was defined were the backscatter coefficients reach*

*the molecular background and the lower boundary of the lofted layer was set to the minimum in the backscatter coefficient profile between the PBL and the lofted layer. We added this point as follows:*

The top heights and the depths of the lofted aerosol layers are shown in Fig.5. Both values were identified visually using the backscatter coefficient profiles. The top height was defined were the backscatter coefficients reach the molecular background and the lower boundary of the lofted layer was set to the minimum in the backscatter coefficient profile between the PBL and the lofted layer.

7. In the Section 4.2 Aerosol classification, In Fig.7, are the data points the daily averaged values? Those circles marked for the aerosol types seem arbitrary or not objective based on some thresholds of aerosol optical parameters. What are your methods or any thresholds of aerosol optical properties for classifying these aerosol types? For the given type of aerosols, what is the difference between the "Pollution" and "Pollution mixture" aerosol? "Burning product" is a little confused, "biomass burning"?

- *Also here the data points are the same values for the 2-3 hours averaged profiles. The classification is based on the values obtained from lidar observations during recent years, especially in the frame of EARLINET and PollyNET. The respective literature is cited in the discussion of these Figures.*

- *Pollution mixtures in contrast to pollution refers to depolarisation ratios between 5% and 10%. These are caused by larger or more spherical particles. This aerosol type is discussed on page 9 line 6-10 in the chapter 4.2.*

- *Burning products include particles from biomass burning, industrial burning or domestic burning. We replaced the expression by "particles from burning processes" at two incidences.*

  *Page 9, line 22 : … smaller particles from burning processes that contain soot.*
  *Page 9, line 29: … for urban particles and particles arising from burning processes.*

8. In the Section 4.3 Origin of the aerosol layers- trajectory analysis, The lofted aerosols below 1200-m are probably from the local nocturnal residual layer since they are so low or in the PBL, thus they are probably not from the long-range transport.

- *Yes that is right. Most of the observed aerosol layers origins from local sources. This is part of the results of the trajectory analysis on page.*

Page-10, Line-12, a total number of 413 backward trajectories was obtained. It seems that they are not the daily averaged profiles since your total observation days are less than this number. How long is the time average for a lidar profile? That means that on some day you might have a lot of aerosol profiles while on other days you might only have one or none.

- *The time average for each profile is always 2-3 hours. The total number of profiles used is the same as before, but some profiles show more layers, so that the total number of upper-layers used was 147. Trajectories were calculated for three, sometimes five altitudes for each profile. Thus, the total number of 413 trajectories arises. Also here, the maximum number of profiles per day was restricted to four (see manuscript page 4, line 1-2)*

9. In the Section 4.4, If possible, the statistics of PBL aerosols optical properties can be given for the comparisons with the aloft aerosols because the PBL aerosol pollutants are more related to the human health or draw more attentions.

- *Due to the incomplete overlap, we cannot evaluate the particle extinction coefficient inside the PBL (see also answers to 1 and 4 above). This affects also the lidar ratios at 355 and 532 nm and the extinction Angström exponent. The depolarisation ratio is always below 5%, so only the statistics of the backscatter Angström exponent are left. Thus, another plot would not give much information. We prefer to concentrate this study on the lofted aerosol layer.*

10. In the Section 5 Conclusion, Page-12, Line 22-23, authors mention, "This was also observed in Guangzhou and is consistent with the Asian monsoon circulation in the region." There are no enough discussions about the effects of Asian monsoon circulation on the aerosols. How does the Asian monsoon affect the aerosols?

- *We were referring to the wind direction that is following with the Asian monsoon. This might not be made clear here. New part of the sentence:*

…and is consistent with the general wind circulation dominated by the Asian monsoon.

Page-12, Line 26-27, "The particles are locally and regionally produced and are only seldom mixed with transported particles from further away." This is not consistent with the Figure 8 (b) and (c), even Figure 8 (a). For instance, in Fig.8 (b), the cluster-3 for the long-distance transport shows 31% percentage against the 38% of the Cluster-1.

- *Long calculation time of the trajectories does not exclude that the aerosols come from sources close by. Our statement is based on the optical properties that were observed by the lidar measurements. We have adapted the text accordingly:*

These particles are mainly locally and regionally produced. During the summer monsoon season, they may also be mixed with particles of marine origin from the close-by sea. Dust mixtures into the pollution aerosol transported from sources further away was only observed in one case.

Technical corrections:

1. Page-1, Line-6, two "observed" appear in the sentence. Please delete the first one and move "by the sunphotometer" afterward to the second "observed". Please give the wavelength for the aerosol optical depth and lidar-ratio. - *done*

2. Page-1, Line-8, please delete the word "even". - *done*

3. Page-1, Line-9, "aerosol" should be "aerosol types". - *done*

4. Page-1, Line-11, please add "%" behind the number "3.7". - *done*

5. Page-1, Line-12, you may say the mixture of fine and coarse-mode aerosols. - *done*

6. Page-1, Line-13, the word "mainly" should be "main".

– *We wanted to express what is 'most of the time' = 'mostly' present in the atmosphere above PRD. We replaced 'mainly' by 'mostly'.*

7. Page-2, Line-9, please add the word "for" in front of "most of the time in the PRD". *- done*

8. Page-3, Line-21, please revise the word "is increasing" with "increases".

 *- We changed "is including" with "includes" -We did not find "is increasing"*

9. Page-3, Line-26, please delete the word "also". *- done*

10. Page-4, Line-1, this sentence is confused.

*– We changed the sentence:*
To avoid over-representation of long lasting cloud-free periods with constant aerosol conditions, the number of considered profiles per day during such periods was reduced to a maximum number of four.

11. Page-6, Line-22, "04:30 h" should be "04:30 am". *- done*

12. Page-8, Line-2, please revise the sentence or just say:
"The top heights of the lofted aerosol layers range from a few cases of 1.5 km to 5 km (Fig. 6)." *- done*

13. Page-8, Line-9, please delete the word "depths" after " 3 km". *- done*

14. Page-8, Line-13, the word "is" should be 'are". *- done*

15. Page-9, Line-30, the word "6%" should be "6 sr". *- done*

16. Page-11, Line-11, the word "calculate" should be "calculated". *- done*

17. Page-11, Line-22, please add "sr" behind the number "50.7". *- done*

18. Page-12, Line-7, the number "042" should be "0.42". *- done*

**List of other relevant changes**

1) Several occurrences of the word "depolarisation" are replaced by "depolarization".

They are all marked in the following marked manuscript.

2) I used LaTeXdiff to produce this marked manuscript to follow changes, as recommended. The program did mark all the changes correctly, but it also stopped at some point (after page 14). I could not find the reason for this, because LaTeXdiff introduced 68 new compilation errors, mainly pointing to some keyboard character problems. I guess there was some interference with characters in the text and LateXdiff, which I could not solve easily and fast. Therefore, I added the remaining pages (15 - 26) of the plain revised manuscript and highlighted the changes with a yellow marker. These are mainly figures and tables, so that there were only a few changes left.

The resulting manuscript is thus a mixture of LaTeXdiff produced pagers and normal pages. I hope this combination of pages can be accepted as the requested marked manuscript.

[revised manuscript text omitted]

**Figure 3.** Lidar optical property profiles of particle backscatter coefficient (BSC), particle extinction coefficient (EXT), lidar ratio, Ångström exponent and linear particle depolarization ratio at 532 nm derived from Polly$^{XT}$ measurement on March 26, 2012 (local time is + 8 h, 02:00 to 04:30 LT)). The profiles are vertically smoothed over 15 rangebins which corresponds to 450 m. The dashed prolongation of the particle extinction profiles towards the ground result from fitting the BSC to the EXT profile and extrapolation to the ground.

[Figure]

**Figure 4.** Trajectory analysis for March 26, 2012, 48 h back in time, arriving at 500 m inside the boundary layer, at 2200 meter below the lofted layer, and at 3000 m inside the lofted layer.

[Figure]

**Figure 5.** All single and seasonal mean particle extinction coefficient profiles measured the by the lidar at 532 nm during the entire observation period from November 2011 to June 2012.

[Figure]

**Figure 6.** Histogram of lofted layer heights and depths

[Figure]

**Figure 7.** Aerosol classification using lidar ratio at 532 nm versus linear particle depolarization (left panel) and Ångström exponent for the particle backscatter coefficient for 532 nm / 1064 nm (right panel) for lofted layers in the period from November 2011 to June 2012. The colored circles indicate the identified particle mixtures and corresponding measurements.

[Figure]

Cluster means - Standard   GDAS Meteorological Data
124 backward trajectories

Layer top > 3500 m

3 ( 28%)
2 ( 35%)
1 ( 37%)

c)

Source at 23.06 N; 113.39 E

268 backward trajectories

Layer top > 1200 m

4 ( 2%)
2 ( 23%)
5 ( 7%)   3 ( 31%)
1 ( 38%)

b)

[Figure]

146 backward trajectories

Layer top < 1200 m

5 ( 9%)
4 ( 10%)
3 ( 15%)
1 ( 37%)
2 ( 29%)

a)

**Figure 8.** Trajectory cluster means for (a) PBL top heights up to 1200 m, (b) all lofted layers above 1200 m, and (c) lofted layers top height above 3500 m. All trajectories were calculated 144 h back in time.

[Figure]

**Figure 9.** Statistical analysis of the optical properties of the observed lofted aerosol layers: (a) Lidar ratio at 355 nm, (b) Lidar ratio at 532 nm, (c) linear particle depolarization ratio at 532 nm, (d) Extinction Ångström exponent, (e) Backscatter Ångström exponent (355 nm / 532 nm), (f) Backscatter Ångström exponent (532 nm / 1064 nm).

---

## Author Response (AR2)

**Answer to the Co-Editor**

Comments to the Author:
The authors have replied the reviewers' comments adequately in most cases. However, the paper still contains a large number of typos, awkward English (e.g. frequent switching of time between past tense and present tense), etc. This needs to be improved before the paper can be further evaluated. See examples of corrections (not intended to be complete) in the attached file.

*Answer:*
*We thank the Co-Editor for his careful reading of the manuscript and have corrected all incidences of past- and present tense as well as all typos. In addition we changed a few sentences regarding their syntax.*

*Please find all changes made in the following marked up document produced by Latex Diff. Again, the Latex Diff software did not find the right figures and figures numbers. However, all changes made are indicated. The figure numbers are correctly displayed in the revised manuscript file.*

[revised manuscript text omitted]